# Paclitaxel-Trastuzumab Mixed Nanovehicle to Target HER2-Overexpressing Tumors

**DOI:** 10.3390/nano9070948

**Published:** 2019-06-29

**Authors:** Celia Nieto, Ariana Centa, Jesús A. Rodríguez-Rodríguez, Atanasio Pandiella, Eva M. Martín del Valle

**Affiliations:** 1Departamento de Ingeniería Química y Textil, Facultad de Ciencias Químicas, Universidad de Salamanca, 37008 Salamanca, Spain; 2Instituto de Biología Molecular y Celular del Cáncer (IBMCC), CSIC, CIBERONC-IBSAL, 37007 Salamanca, Spain

**Keywords:** targeted nanoparticles, paclitaxel, trastuzumab, HER2-specificity, sodium alginate, piperazine

## Abstract

Paclitaxel is one of the most widely used chemotherapeutic agents thanks to its effectiveness and broad spectrum of antitumor activity. However, it has a very poor aqueous solubility and a limited specificity. To solve these handicaps, a novel paclitaxel-trastuzumab targeted transport nanosystem has been developed and characterized in this work to specifically treat cancer cells that overexpress the human epidermal growth factor receptor-2 (HER2). Methods: Alginate and piperazine nanoparticles were synthetized and conjugated with paclitaxel:β-cyclodextrins complexes and trastuzumab. Conjugated nanoparticles (300 nm) were characterized and their internalization in HER2-overexpressing tumor cells was analyzed by immunofluorescence. Its specific antitumor activity was studied *in vitro* using human cell lines with different levels of HER2-expression. Results: In comparison with free paclitaxel:β-cyclodextrins complexes, the developed conjugated nanovehicle presented specificity for the treatment of HER2-overpressing cells, in which it was internalized by endocytosis. Conclusions: It seems that potentially avoiding the conventional adverse effects of paclitaxel treatment could be possible with the use of the proposed mixed nanovehicle, which improves its bioavailability and targets HER2-positive cancer cells.

## 1. Introduction

Cancer therapy improvement has become one of the most important health challenges for the time being due to the high incidence of this disease. Systemic chemotherapy is the necessary choice when cancer is late diagnosed and, unfortunately, it is often accompanied by severe side effects [1,2]. As most employed cytotoxic agents affect cell cycle progression, not only cancer cells are killed by antitumor drugs, but also normal cells that proliferate rapidly [3,4].

Such side effects are present, for instance, in paclitaxel (PTX)-containing treatments. This taxane diterpenoid drug targets β-tubulin in cells and arrests them in the G2/M cell cycle phase [5]. Due to its broad spectrum of antitumor activity and high effectiveness, it is normally employed to treat breast, ovarian, and non-small cell lung cancers [6]. In this manner, PTX is one of the three most administered chemotherapy agents today, but it presents an important handicap. Its aqueous solubility is very limited (0.3–0.5 μg/mL) and intravenous PTX’s formulations have had to be developed [7]. Among them, they are only two that are approved to be used in clinic and the major drawback is that, in the most used one [5], PTX is dissolved into a chemical solvent mixture that entails important side effects. Thus, such mixture reduces the taxane effectiveness and limits the doses that can be administered to patients [6].

In this context, the need of novel PTX’s therapeutic systems and the relevance of the nanomedicine field in this task are understood. Nowadays, PTX inclusion complexes with cyclodextrins as well as PTX polymeric nanoparticles, liposomes, micelles, and polymeric conjugates can be found in the literature [6,8,9,10,11]. All of them look for an improved vehiculization of the taxane and this fact can be achieved by the addition of a specific monoclonal antibody (mAb) to the nanosystems. mAbs are excellent targeting proteins and their inclusion in drug delivery vehicles could enhance their endocytosis and lysosomal degradation, with the resulting release of the cytotoxic agent [12].

One humanized mAb approved by the U.S. Food and Drug Administration (FDA) is trastuzumab. It binds to the extracellular domain of the HER2/neu receptor, inhibits downstream signaling and activates immune mechanisms that ultimately contribute to its antitumor effect [13]. HER2 is overexpressed in a variety of carcinomas, like breast, ovarian, lung, and gastric tumors [14,15]. Among them, trastuzumab administration is allowed to treat HER2-positive metastatic breast cancers and HER2-positive stomach tumors and it has transformed the complicated prognosis of these subtypes of carcinomas [16]. Nevertheless, although this mAb has demonstrated a proper antitumor efficacy, it is normally administered in combination with chemotherapeutic drugs, like taxanes, to enhance the effect of both of them [14,17]. However, one of the limitations of trastuzumab-taxane therapies is their toxicity.

Given everything here mentioned, the objective of the present work has been the development, characterization, and validation of a novel administration PTX’s nanovehicle, consisting of sodium alginate and piperazine nanoparticles (APPZ) [18]. In such nanosystem, PTX has been included into β-cyclodextrins molecules (βCD) to improve its water-solubility. Moreover, trastuzumab has also been attached to APPZ surface. The aim of the addition of this antibody was to favor specific targeting of HER2-overexpressing cancer cells.

Of the chemical compounds named above, sodium alginate is a harmless anionic polysaccharide that is commonly used as a gelling agent for cosmetic and food products [19]. As for piperazine, it is a heterocyclic compound having two nitrogen atoms at opposite positions in a six membered ring that confer it a potent antioxidant power [20]. Piperazine derivatives have been developed in recent years as a basis for alternative antitumor agents and, in our case, such chemical compound allowed a chemical crosslinking of the sodium alginate chains and, thus, the formation of nanoparticles [18,21]. Finally, βCD are cyclic oligosaccharides with a hydrophobic interior. These molecules are able to form inclusion complexes with a variety of lipophilic molecules and are used for many pharmaceutical applications [22]. Besides, βCD have become commonly employed in nanoparticle-based drug delivery systems thanks to their benefits [23,24,25] and some nanovehicles based on βCD-PTX complexes have been already describe in literature [26,27].

## 2. Materials and Methods 

### 2.1. APPZ Synthesis and Characterization

APPZ were synthesized by preparing sodium alginate (1 mg/mL) and piperazine (2 mg/mL) solutions with a 4.7 pH value [18]. Different amounts of the piperazine solution were dropped over the alginate one. The mixture was magnetically stirred and suspensions of APPZ with different properties were obtained. To isolate them, pH of the suspensions was dropped to 1.0 and APPZ were centrifuged.

In order to characterize them, APPZ were re-suspended in deionized water (1 mg/mL). The pH of these suspensions was set again at 4.7. APPZ sizes and zeta potentials were measured by Dynamic Light Scattering (DLS) and Laser Doppler Electrophoresis (LDE) with a 90˚ fixed angle detector, using a 633 nm wavelength laser (ZetaSizer Nano ZS90, Malvern Instruments Inc., Royston, Hertfordshire, UK). APPZ with the best size/surface charge values were selected and pH effect over both parameters was studied with them. Variation of their diameter and zeta potential was analyzed over 8 consecutive days to determine APPZ stability over time. In addition, such properties were also studied at pH 7.2, re-suspending selected APPZ in PBS.

### 2.2. Preparation of PTX:βCD Inclusion Complexes

PTX:βCD complexes were prepared by means of a freeze-dying method. PTX was dissolved in pure ethanol (0.8 mg/mL) in dark conditions, whereas βCD were dissolved in deionized water (0.9 mg/mL). βCD aqueous solution was added to the PTX one (1:1 molar ratio) and the resulting hydroalcoholic solution was mechanically stirred for 5 hours. Then, it was freeze-dried and dissolved in deionized water [28,29]. PTX inclusion efficiency in βCD was analyzed by mass spectrometry (LC 2795 chromatograph connected to a ZQ-4000 mass spectrometer, Waters Corporation, Milford, MA, USA) [30].

### 2.3. APPZ Conjugation with Trastuzumab and PTX:βCD Complexes and Conjugated APPZ Characterization

1-Ethyl-3-(3-dimethylaminopropyl)carbodiimide/N-hydroxysuccinimide (EDC/NHS) coupling chemistry was the one selected to conjugate APPZ with trastuzumab and the obtained PTX:βCD complexes. EDC (193 mg/mL) and NHS (58 mg/mL) aqueous solutions were prepared and added to an APPZ suspension (100 mg/mL, pH 4.7). The mixture was stirred for 40 minutes and a trastuzumab (3.2 µg/mL) aqueous solution was later joined. The suspension was again stirred for 3 h and PTX:βCD complexes (0.8 mg/mL PTX equivalent) were added to it. Final suspension was kept in agitation in the dark overnight [31,32,33,34]. Conjugated APPZ were isolated by decreasing suspension pH value until 1.0 and by centrifuging them [18]. Supernatant was preserved to determine PTX:βCD and trastuzumab conjugation efficiencies (CEs) on APPZ.

To finish, conjugated APPZs were characterized. Their size and zeta potential were measured at the conjugation and at the physiological pH values in deionized water and phosphate buffered saline (PBS), respectively. Moreover, these parameters were determined over 8 consecutive days at pH 4.7 to study conjugated APPZ stability, too.

### 2.4. Determination of PTX and Trastuzumab Conjugation Efficiencies on APPZ 

The amount of PTX:βCD and trastuzumab present in APPZ supernatant were analyzed by mass spectrometry and the Bradford method, respectively. In this last case, APPZ supernatant (100 µL) was mixed with Bradford reagent (1 mL). After 5 min, the absorbance of the mixture was measured at 595 nm (UV-1800 spectrophotometer, Shimadzu Corporation, Soraku-gun, Kyoto, Japan) [35]. Once trastuzumab amount present in APPZ supernatant was known, the antibody and the drug CEs were determined by difference, according to equation (1):

CE = (PTX/trastuzumab initial amount − supernatant PTX/trastuzumab)/nanoparticles amount (1)

### 2.5. Cell Culture

BT474, SKBR3, OVCAR3, and HS5 cell lines were cultured with medium supplemented with FBS (10%), containing high glucose (4500 mg/L) and antibiotics (penicillin 100 U/ml, streptomycin 100 mg/mL). Cell lines were cultured at 37 °C in a humidified atmosphere in the presence of CO_2_ (5%).

### 2.6. Determination of BT474, SKBR3, OVCAR3, and HS5 Cellular Levels of HER2-Expression

To determine HER2 and phosphorylated-HER2 (pHER2)-expression in the BT474, SKBR3, OVCAR3 and HS5 cell lines a western blot was carried out. Calnexin was selected as a loading control. Detailed procedures for the protein extraction, quantification, and immunoprecipitation, as well as for the western blotting, can be consulted in Montero et al. [36].

### 2.7. Conjugated APPZ Internalization in HER2-Overexpressing Cancer Cells

BT474 cells were cultured on coverslips and incubated with culture medium supplemented with FBS (10%) and chloroquine (50 µM) for 1 h. Cells were treated with conjugated APPZ (6.8 μg PTX-equivalent/mL medium) for 30 minutes, 4, 24, and 48 h. Dilutions (1:100) of the anti-LAMP1 antibody were employed for analyzing the HER2 and cellular lysosomes co-localization.

After conjugated APPZ treatment, an immunofluorescence assay was performed according as it was described in Esparís-Ogando et al. and analyzed by confocal laser scanning microscopy (CLSM) (Leica TCS SP5, Leica Microsystems, L’Hospitalet de Llobregat, Spain) [37].

### 2.8. Conjugated APPZ Specificity: Targeting HER2-Positive Cancer Cells

A BT474 and HS5 cell lines’ co-culture was treated with conjugated APPZ in order to demonstrate their specificity. BT474 and HS5 cell lines were stained with CellTracker^TM^ Green CMFDA and CellTracker^TM^ Red CMPTX, respectively. After 24 h, both cell lines were seed together in 6-well plates in a 150,000:300,000 BT474:HS5 cellular ratio. Next day, cells were treated with PTX:βCD complexes (6.8 µg PTX-equivalent/mL) and conjugated APPZ (6.8 μg PTX-equivalent/mL) for 48 h. The effect of both systems was analyzed each 24 h by immunofluorescence [37]. In order to compare the specificity of the free PTX:βCD complexes and the conjugated APPZ, the number of cells of the BT474 and HS5 cell lines was counted in 12 different images for each treatment and the mean ± SD was calculated.

### 2.9. Conjugated APPZ Cytotoxic Effect According to Cellular Levels of HER2-Expression

Antitumor activity of PTX:βCD complexes and trastuzumab conjugated APPZ (APPZ-PTX:βCD-T) was studied through 3-(4,5-Dimethylthiazol-2-yl)-2,5-Diphenyltetrazolium Bromide (MTT) assays [38]. Cells from the BT474, SKBR3, OVSCAR3, and HS5 cell lines were seed in 24 well-plates and cultured with medium, supplemented with FBS (10%), overnight. The next day, culture medium was replaced by supplemented medium containing PBS (control), free PTX:βCD complexes (6.8 µg PTX-equivalent/mL), trastuzumab (T) (8 ng/mL), APPZ (1 mg/mL), and APPZ-PTX:βCD-T (1 mg/mL, 6.8 μg PTX-equivalent/mL). Free PTX (6.8 µg/mL), previously dissolved in PBS, and APPZ-PTX:βCD (1 mg/mL, 6.8 μg PTX-equivalent/mL) were also tested in the BT474 and HS5 cell lines. In all cases, cell proliferation was analyzed for 4 days, each 24 h. Shown results are the mean ± SD of four replicas for each different treatment.

### 2.10. Statistical Analyses

Data shown in APPZ characterization, regarding to their size and zeta potential, is the mean ± SD of three different measurements. Results of the MTT assays have been represented as the mean ± SD of four replicas for each treatment of three different experiments and results were considered statistically significant were *p* < 0.05. Otherwise, the quantification of the bands obtained in the western blot assay was carried out with the ImageJ 1.44 software (National Institutes of Health, Bethesda, MD, USA). The intensity of each band was determined with regard to the control values and data has been represented as the percentage of the maximum value obtained for each experiment.

## 3. Results

### 3.1. Stable APPZ Synthesis and Characterization

As it was mentioned before, APPZ were synthesized by dropping a piperazine aqueous solution over a sodium alginate one, in different molar ratios, at pH 4.7. Mannuronic and guluronic acid units from sodium alginate chains have a pKa of 3.4 and 3.7, respectively [39]. Piperazine pKa values are, approximately, 5.7 and 9.7 [40]. In this manner, at the working pH, electrostatic interactions took place between both compounds and nanoparticles were formed due to a sodium alginate chains’ chemical crosslinking (Figure 1a [18]).

Once obtained, APPZ were isolated, re-suspended in deionized water and characterized. Size and zeta potential of the APPZ synthetized with different piperazine:alginate molar ratios were determined (Appendix A). The best size/surface charge relationship was achieved with a 0.25 piperazine:alginate molar ratio and APPZ developed in this manner were selected for further characterization. First, their size and superficial charge were determined at the synthesis pH, finding values of 208 ± 40 nm (PDI = 0.3) and −13.2 ± 1.6 mV, respectively (Figure 1b). Next, pH influence over both parameters was analyzed (Appendix A). As it is shown, APPZ size did not vary significantly at pH values between 3.0 and 6.0. However, it increased due to the loss of the electrostatic interactions between the alginate chains and piperazine at very acid or basic pH values. As for the APPZ zeta potential, it had more positive values at acid pH values due to the protonation of the –COOH alginate groups. APPZ stability over time was also studied. Variation of their diameter and zeta potential was determined over 8 consecutive days, preserving APPZ at room temperature (Figure 1c).Over this period, size only varied a 3.5% ± 0.27% in respect of the first measurement obtained, while superficial charge changed a 5.4 ± 0.3% among the different days of the study. Finally, APPZ properties were analyzed when they were re-suspended in PBS at pH 7.2 and, in this case, they showed a 214 ± 55 nm mean diameter (PDI = 0.4) and a −19.8 ± 1.1 mV superficial charge.

### 3.2. APPZ Conjugation with Trastuzumab and PTX:βCD Complexes and Later Characterization

Before starting with APPZ conjugation, PTX inclusion efficiency in βCD was analyzed by High-Performance Liquid Chromatography (HPLC) and mass spectrometry. It was found that such inclusion efficiency was about a 0.79 mg PTX/mg βCD [30].

Then, once APPZ were isolated after being synthetized, trastuzumab and the obtained PTX:βCD complexes were covalently attached to their surface (Figure 2a). EDC, in a non-cytotoxic concentration (0.01M) [34], was employed to transform alginate carboxylic groups into stable O-acylurea intermediates. Besides, NHS was added to improve EDC stability [33]. Later, O-acylurea intermediates reacted with trastuzumab amine groups from lysine residues to form amide bonds when the antibody was joined [12,14]. In the same manner, PTX:βCD complexes were added to the active APPZ suspension and βCD molecules were attached to them through ester bonds.

Non-attached trastuzumab and PTX:βCD complexes were discarded by a second APPZ isolation through centrifugation. The amount of the antibody and the antitumor drug present in conjugated APPZ supernatant was analyzed. CEs were found to be 8 ng trastuzumab/mg APPZ and a 6.8 μg PTX/mg APPZ, which were adequate results according to the doses of these compounds that have to be administered in vivo [41,42]. Likewise, trastuzumab and PTX:βCD complexes attachment to APPZ was proved with a Fourier Transform-Infrared Spectroscopy (FT-IR) analysis in the 500–4000 cm^−1^ region (Figure 2c). Compared with the APPZ, the main changes in the APPZ-PTX:βCD-T spectra appeared at 1232, 1704, and 3500–4000 cm^−1^. The first shift could correspond to the βCD conjugation through an ester bond, while the other two demonstrated trastuzumab attachment. Thus, the shift from 1704 cm^−1^ could be assigned to an amide bond vibration and there could be more amide shifts in the 1570–1670 cm^−1^ region that would be masked by the wide alginate signal [34].

To finish with conjugated APPZ characterization, their size and zeta potential were determined. At the conjugation pH value, their size was near 296 ± 50 nm (PDI = 0.4), while their superficial charge was about −24 ± 1.4 mV (Appendix A). Otherwise, when they were re-suspended in PBS, the values of these parameters were 359 ± 60 nm (PDI = 0.45) and –26.2 ± 2.1 mV at the physiological pH (7.2). Conjugated APPZ stability was also studied for 8 consecutive days at pH 4.7 (Figure 2b). Variations in their diameter and superficial charge were determined and it was found that first property varied a 2.25% ± 0.84 % in respect of the first day measurement. On the other hand, superficial charge changed a 3.13% ± 0.27 % among the different performed measurements.

### 3.3. Conjugated APPZ are Internalized in HER2-Overexpressing Cancer Cells by Endocytosis

Different cell lines expressing distinct HER2 levels were selected to study conjugated APPZ biological properties. As described in the literature [43,44], it was found that total HER2 and pHER2 were great overexpressed in the BT474 and SKBR3 cell lines. OVCAR3 and HS5 cells also expressed HER2, although to a much lesser extent. This fact can be noticed in the overexposed western blot (Figure 3a).

Conjugated APPZ internalization was studied in BT474 cells. Thanks to the presence of trastuzumab in the APPZ surface, it was expected that APPZ would be internalized in HER2-overpressing cells through an endocytosis process. In addition, it was also considered that part of the administered PTX could pass through the cellular membrane and arrived directly to the microtubules, its therapeutic center of action (Figure 3b). The first fact was proven by means of immunofluorescence and CLSM. BT474 cells were incubated with conjugated APPZ for different times and the location of trastuzumab was visualized using a secondary antibody. Incubation with APPZ resulted in progressive accumulation of trastuzumab in intracellular sites (Figure 3c). Furthermore, a co-staining of trastuzumab and the cellular lysosomes was carried out, using a LAMP1 marker, to check whether APPZ were targeted to lysosomes. There was a co-localization of both markers that proved that APPZ would end up in such organelles in treated cells (Figure 3d).

### 3.4. Conjugated APPZ Present HER2-Overexpressing Specificity

In order to compare conjugated APPZ specificity with that of the free PTX:βCD complexes, a co-culture of the HER2-overexpressing BT474 and the human stromal HS5 cell lines was performed. Both cell lines were separately stained and once they were cultured together, they were treated with conjugated APPZ (Figure 4a). CLSM images were taken 24 and 48 h after such treatment (Figure 4b). It could be seen that free PTX:βCD complexes really affect to BT474 cells’ viability, but they also notably reduced HS5 cells’ survival, especially after 48 h. By contrast, conjugated APPZ achieved to reduce BT474 cells’ viability without affecting HS5 cells’ survival in an analogous manner (Figure 4c).

### 3.5. Conjugated APPZ Have Different Cytotoxic Effect According to Cellular Levels of HER2-Expression

Conjugated APPZ antiproliferative activity was analyzed through MTT assays with the BT474, SKBR3, OVCAR3, and HS5 cell lines. As it was shown before, BT474 and SKBR3 cells presented a HER2-overexpression, while such expression was lower in the OVCAR3 and HS5 cell lines, respectively (Figure 3a). Thus, conjugated APPZ resulted almost as efficient as equivalent free PTX:βCD complexes in the first two HER2-overpressing cell lines (Figure 5a,b). Nevertheless, their antitumor effect decreased in the non HER2-overexpressing cells in comparison with the free PTX:βCD complexes activity (Figure 5c,d). This difference proved again the specificity of the targeted nanosystem. In addition, it could be seen that non-conjugated APPZ decreased tumor cells’ proliferation rate, possible due to piperazine antioxidant effect [45], but that such treatment did not affect non-HER2-overexpressing cells’ survival rate.

With the aim of demonstrating that the specificity of the targeted nanovehicle was favored by the conjugation of trastuzumab, APPZ-PTX:βCD effect over cellular viability was also tested in BT474 and HS5 cells. As result, it was proven that APPZ-PTX:βCD decreased BT474 cells’ survival rate in a lesser extent than APPZ-PTX:βCD-T did, probably because of a minor endocytosis rate, non-receptor mediated. On the contrary, APPZ-PTX:βCD were responsible for a further reduction in HS5 cells’ viability in comparison with APPZ-PTX:βCD-T (Appendix A).

At last, improvement of PTX aqueous solubility with its inclusion into βCD was shown with MTT experiments with BT474 and HS5 cell lines, too. Free PTX was tested in both types of cells and it was not as much efficient as PTX:βCD complexes were. Even conjugated nanoparticles had a more remarkable effect over BT474 cellular viability (Appendix A).

## 4. Discussion

In spite of being one of the most employed chemotherapy drugs these days, PTX has a very limited aqueous solubility. As consequence, it is normally administered with solvents that add more toxicity to the taxane and limit their bioavailability. Besides, PTX inhibits microtubules’ depolymerization and presents a restricted tumor-selectivity and all these handicaps confine its possible given doses [6,7].

Herein, the scientific community is making great efforts in order to reduce these PTX disadvantages and [7,8], in this work, such drawbacks’ reduction was the main aim.

In this manner, a novel PTX targeted nanovehicle, which had been synthetized in previous works to transport other conventional drugs [18], was developed. Integrated by piperazine and sodium alginate nanoparticles, such nanovehicles were obtained thanks to the electrostatic interactions that took place between both biocompatible compounds at acid pH values. Once obtained, the PTX transport nanosystem was characterized at the development pH value. Moreover, their size and superficial charge were determined at other pH values, like the physiological pH. In any case, APPZ showed zeta potential values that guaranteed their colloidal stability and that, along with their mean size values and stability, were adequate for a potential in vivo administration of the nanoparticles [46].

On the developed nanovehicle’s surface, trastuzumab and PTX, included in βCD molecules, were attached thanks to the EDC/NHS chemistry, previously employed in literature to anchor proteins on other nanoparticles and hydrogels [33,34]. PTX was included in βCD with the objective of improving its solubility in aqueous media [22] and the used of trastuzumab was intended to target HER2-overexpressing cells. Their attachment to the APPZ surface was proven with a FT-IR analysis and the CEs of both, the antibody and the PTX:βCD complexes, were suitable according to the doses of these compounds that have to be administered to achieve a therapeutic effect. Once conjugated, APPZ were again characterized and anew presented a proper size, zeta potential, and stability to potentially be in vivo administered [47].

For the next step, conjugated APPZ internalization was studied in the BT474 HER2-overexpressing cell line. As expected [46], it was demonstrated with an immunofluorescence assay and CLSM that conjugated APPZ were internalized into cellular cytoplasm by endocytosis. Thus, it could be seen that trastuzumab fluorescence signal was displaced from the cellular membrane to the cytoplasm only after 4 h of treatment and that such signal overlapped with the one of the lysosomal marker LAMP1 [48].

Finally, conjugated APPZ specificity and antitumor activity was in vitro validated to check if the main objective of the work was accomplished. For the first fact, a co-culture between the BT474 and the HS5 cell lines was carried out and treated with the targeted nanovehicle and with equivalent free PTX:βCD complexes. As result, it was shown that while the free PTX:βCD complexes reduced both the HER2-overexpressing and the non-HER2-overexpressing cellular viability in a notable manner, conjugated APPZ were more specific. They were almost as much efficient as PTX:βCD complexes killing BT474 cells after 48 h, but they did not reduce HS5 viability in the same way. Even the number of such healthy cells was increased in comparison with the control 48 h after being treated with the conjugated APPZ, possibly due to higher space availability in the culture wells.

Furthermore, conjugated APPZ specificity was afresh demonstrated when their antiproliferative activity was analyzed with MTT assays. When HER2-overpressing cell lines (BT474 and SKBR3) were treated with them, conjugated APPZ almost had the same viability effect as the free PTX:βCD complexes. However, in the cell lines that had a lower HER2-expression (OVCAR3 and HS5), there was a significant difference between the targeted nanosystem and the PTX:βCD complexes activity. Hence, it was shown that PTX aqueous solubility, bioavailability, and specificity conferred by the presence of trastuzumab, were improved and that the aim of the study was attained.

## 5. Conclusions

On the whole, a novel and targeted PTX nanovehicle has been developed in this work with the aim of improving its bioavailability and specificity for the treatment of HER2-positive tumors. Integrated by piperazine and sodium alginate nanoparticles, the surface of such nanovehicles was properly activated by means of the EDC/NHS chemistry to covalently attach to it trastuzumab and PTX, included in β-cyclodextrin molecules to improve its water-solubility.

Once conjugated, the nanosystem showed a proper size, surface charge, and stability to be potentially in vivo administered. In vitro, it was demonstrated that the targeted nanovehicle was internalized into the cytoplasm of HER2-overexpressing tumor cells through an endocytosis mechanism and that it ended up entering their lysosomes. Moreover, the specificity of the conjugated nanoparticles in comparison with that of the free drug was twice proven. Firstly, a co-culture of HER2- and non-HER2-overexpressing cells was performed and, secondly, MTT assays with cell lines with a different level of HER2-expression were carried out. In both cases, the obtained results were similar. While the conjugated nanoparticles were almost as efficient as the free administered PTX:βCD in reducing HER2-overexpressing cancer cells’ viability, they did not affect non-HER2-overexpressing survival rate in the same manner. Such specificity was consequence of the conjugation of trastuzumab to the nanosystem, as additional MTT assays showed. Hence, the employment of the PTX-trastuzumab nanovehicle that is here proposed could help to reduce its frequent adverse effects, being more specific and efficient for the treatment of any HER2-positive tumor.

## Figures and Tables

**Figure 1 nanomaterials-09-00948-f001:**
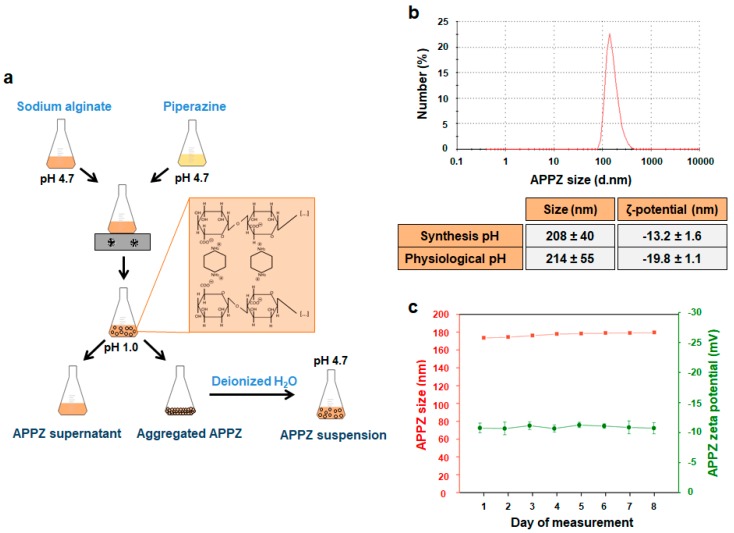
APPZ synthesis and characterization. (**a**) APPZ development. (**b**) APPZ size and a zeta potential at the synthesis (4.7) and physiological (7.2) pH values in deionized water and PBS, respectively. These parameters were determined by DLS and LDE as the mean ± SD of three measurements. The diagram corresponds to APPZ mean size at pH 4.7. (**c**) APPZ diameter and surface charge stability over 8 consecutive days. All the values are the mean ± SD of three measurements, too.

**Figure 2 nanomaterials-09-00948-f002:**
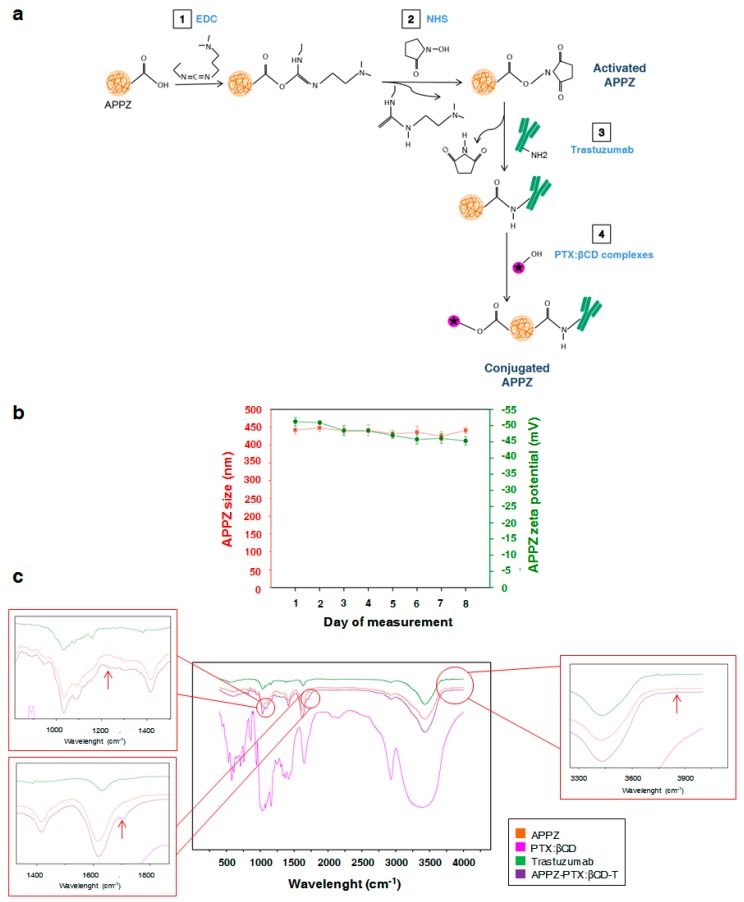
APPZ conjugation and later characterization. (**a**) APPZ conjugation with trastuzumab and PTX:βCD complexes. (**b**) Conjugated APPZ diameter and surface charge stability over 8 consecutive days. All the values are the mean ± SD of three measurements. (**c**) Results of the FT-IR analysis that corroborate APPZ conjugation.

**Figure 3 nanomaterials-09-00948-f003:**
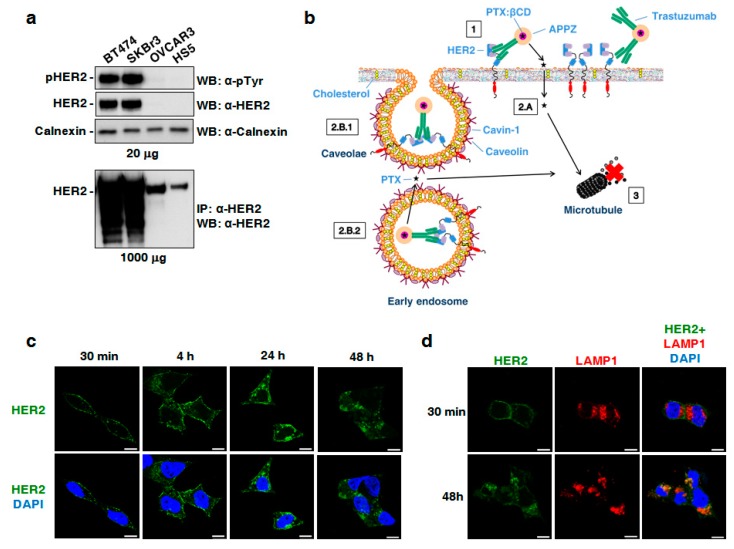
Conjugated APPZ internalization in BT474 cells. (**a**) Study of the HER2 and pHER2 expression levels in the BT474, SKBR3, OVCAR3, and HS5 cell lines. Calnexin was employed as loading control. (**b**) Expected conjugated APPZ internalization into HER2-overexpressing cancer cells. (**c**) HER2 internalization into BT474 cells’ cytoplasm after treatment with conjugated APPZ. Scale bar, 10 µm. (**d**) Co-localization (orange) between HER2 and LAMP1 staining after conjugated APPZ treatment. Scale bar, 10 µm.

**Figure 4 nanomaterials-09-00948-f004:**
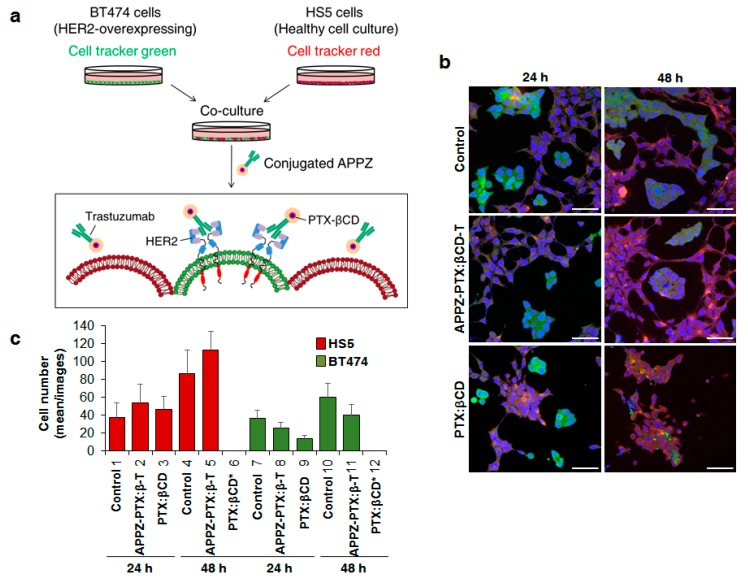
Targeting HER2-overexpressing cells with conjugated APPZ. (**a**) BT474 and HS5 cells were co-culture and treat equivalent free PTX:βCD complexes and conjugated APPZ. (**b**) Obtained results after the treatment of the co-cultured BT474 (green) and HS5 (red) cells, stained with DAPI (blue). Scale bar, 50 µm. (**c**) Living BT474 and HS5 cells from 12 different confocal images were counted in order to analyze the specificity of the conjugated APPZ. Results are the mean of these 12 images ± SD.

**Figure 5 nanomaterials-09-00948-f005:**
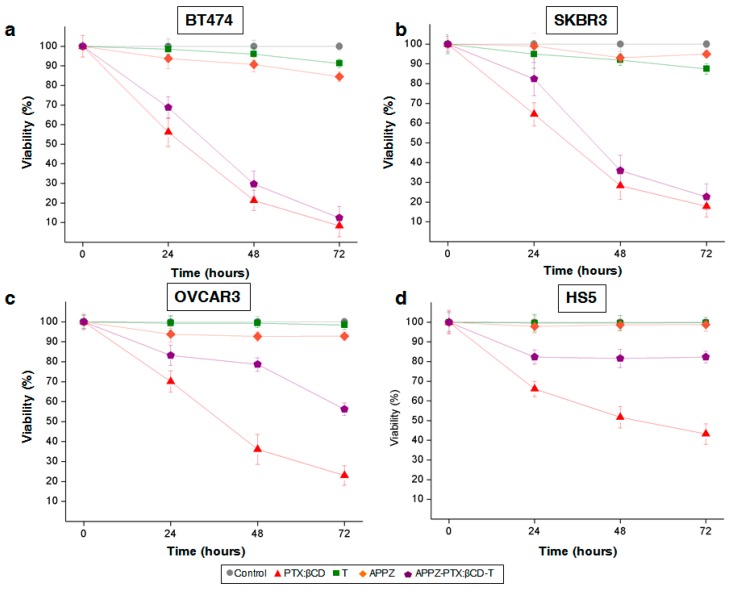
Conjugated APPZ antiproliferation effect in (**a**) BT474; (**b**) SKBR3; (**c**) OVCAR; and (**d**) HS5 cells. Shown results are the mean ± SD of four replicas for each different treatment.

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
