# Peer review of "Paclitaxel-Trastuzumab Mixed Nanovehicle to Target HER2-Overexpressing Tumors"

_nanomaterials, 2019, doi:10.3390/nano9070948_

Reviewer 1 Report

The article is interesting. However, some additional experiments should be performed to support Authors' conclusions.

Authors could decribe in further lenght previous studies using nanovehicles based on beta CD-PTX conjugates such as paper by Minko T et al.

Authors should complete their study with quantitative uptake data such as flow cytometry using fluorescent PTX or HPLC.

Authors should mention size/zeta potentials at physiological pH (7.2-7.4) rather than acidic pH in the text. As the Authors claim in vivo potential of theur nanoparticles, size and zeta potential should be determined in serum-containing media rather than deionized water and, at physiological temperature.

For the MTT experiment, a free PTX group should be included.

A group APPZ-bCD should be included in all experiments to validate the benefit of antibody conjugation.

Author Response

1) Authors could describe in further length previous studies using nanovehicles based on βCD-PTX conjugates, such as paper by Minko T et al.

Previous studies using nanovehicles based on βCD-PTX conjugates have been mentioned at the end of the introduction (lines 81 and 82), and references [26] (Minko et al., 2015) and [27] (Mashru et al., 2018) have been included for such purpose.

2) Authors should complete their study with quantitative uptake data such as flow cytometry using fluorescent PTX or HPLC.

Studying conjugated APPZ internalization in tumor cells with a commercial fluorescent PTX, like the Oregon GreenTM 488 Taxol, was considered. However, we decided to show such internalization employing immunofluorescence and taking CLSM images because, due to the addition of a fluorescent conjugate, the inclusion of the taxane into βCD molecules could be modified. Thus, the conjugation efficiency of the resulting complexes on nanoparticles’ surface could vary and conjugated APPZ endocytosis rate could be smaller due to an increase in nanosystem’s molecular weight. Moreover, it was demonstrated that PTX fluorescent derivatives are less potent than the non-modified drug (Han et al., 1996) and we finally discarded the idea of using them.

Otherwise, we wondered about quantifying PTX uptake in vitro, measuring the amount of the drug present in the culture medium of treated cells. Nevertheless, we realized that the results of this experiment would not guarantee that conjugated APPZ were attached to HER2-overexpressing cells’ surface and/or internalized. In this manner, we determined to employ a fluorescent secondary antibody and a LAMP-1 marker to visually determine the location of trastuzumab, covalently joined to APPZ surface, after cells’ treatment. In any case, if the reviewer considers that quantitative uptake data is necessary, authors will be glad to make the required experiments.

3) Authors should mention size/zeta potentials at physiological pH (7.2-7.4) rather than acidic pH in the text. As the authors claim in vivo potential use of their nanoparticles, size and zeta potential should be determined in serum-containing media rather than deionized water and at physiological temperature.

When MTT and CLSM experiments were carried out, APPZ and conjugated APPZ (APPZ-PTX:βCD-T) were re-suspended in PBS and the suspension pH was set to 7.2. Their size and zeta potential have been measured by DLS and LDE now at these conditions. Thus, APPZ present a mean diameter of 214 ± 55 nm and a superficial charge of -19.8 ± 1.1 mV, while 359 ± 60 nm and -26.2 ± 2.1 mV are the values corresponding to the properties of the conjugated APPZ.

Obtained results have been included in the main text (lines 97 and 108 in the materials and methods and lines 206-207 and 244-246 in results) and Figures 1(B) and S2 have been modified to include the new size and zeta potential values.

4) For the MTT experiment, a free PTX group should be included.

5) A group APPZ-βCD:PTX should be included in all experiments to validate the benefit of antibody conjugation.

4 and 5) In order to answer the two last comments of the first reviewer, new MTT experiments have been performed with an HER2-overexpressing (BT474) and a non HER2-overexpressing (HS5) cell lines. In this novel experiments, PTX, previously dissolved in PBS, and APPZ-PTX:βCD have been tested. With them, it has been demonstrated that trastuzumab conjugation grants HER2-specificity and that PTX aqueous solubility is improved with its inclusion into βCD molecules.

The corresponding methods and results have included in the text (lines 169-171 and 307-306), as well as the Figure S3, shown in the file attached. Moreover, a short statement has been introduced in the discussion and conclusions (lines 373 and 390-391).

These new MTT experiments have only been carried out with two cell lines to have the results in the required revision time period. BT474 and HS5 cells are easier to culture than the SKBR3 and OVCAR3 ones, and authors have chosen them as models of HER2 and non-HER2 overexpressing cell lines to perform the assays. It is though that results are enough to validate PTX aqueous solubility improvement, but authors will be delighted to repeat MTT assays with the OVCAR3 and the SKBR3 cell lines if the reviewer considers this fact necessary and more review time is allowed.

For the same reason, APPZ-PTX:βCD effect has just been tested through the new MTT assays. Authors believe that results are enough to validate the benefit of the antibody conjugation but, as before, they will be happy to performed additional experiments if it would be required by the reviewer and more time is given.

Reviewer 2 Report

The article entitled "Paclitaxel-trastuzumab mixed nanovehicle to target HER2-overexpressing tumors" by Celia Nieto et al.  deals with synthesis and characterisation of Paclitaxel-trastuzumab mixed nanocarriers to target HER2-overexpressing cancer cells. The manuscript is very interesting and well structured. It reports novel findings and conclusions are readily supported by data. The paper requires only minor revision to be accepted for publication in Nanomaterials

Minor issues are reported below:

1) Slight English revision by mother tongue revisor is suggested

2) What about TEM or SEM images of nanoparticles? What about polidisperisity data? Please discuss\update these points

3) Release profiles are missing: slow or fast? Please provide\discuss this point

4) Co-localisation\uptake studies and mechanisms by CLSM or FACS are also overlooked. Please consider these issues

5) References literature could be updated

Author Response

1) Slight English revision by mother tongue revision is suggested.

English revision has been carried out. Several expressions have been changed along the main text (lines 80, 123, 149, 179, 257-258, 260, 267, 285, 297, 302, 337, 342, 349 and 390).

2) What about TEM or SEM images of nanoparticles? What about polydispersity data? Please discuss\update these points.

As far as nanoparticle images are concerned, one of APPZ can be seen in reference [18] (Román et al., 2016). If the reviewer considers that including it in the present work is necessary, authors will be happy to download a formal grant of license and do it. Moreover, images of the conjugated APPZ were taken with environmental scanning electron microscopy (eSEM) at 1˚C in humidified conditions. The SEM-Quanta FEG-250 was employed for such purpose but images were thought to have not enough quality (because of the debris in the back) and were not included in the main text. One of them is shown in the attached file. Nevertheless, if the reviewer believes that it has to become part of one of the figures of the work, authors will be glad to include it in Figure S2.

On the other hand, polydispersity indexes (PDI) of the APPZ and conjugated APPZ have been included in the results of the main text (lines 198 and 243). At the conjugation pH value (4.7), APPZ size presented a PDI = 0.3, while such value was 0.4 for the size of the conjugated APPZ. Also, as the other reviewer has suggested including the size and surface charge of APPZ at the physiological pH, the corresponding PD indexes at this pH have been also introduced in the text (lines 208 and 245).

3) Release profiles are missing: slow or fast? Please provide\discuss this point.

According to literature, due to their size and because conjugated APPZ are a targeted nanosystem, they are expected to be internalized in cells in a different way than free PTX or free PTX:βCD do. Thus, while PTX probably passes through the cellular membrane by diffusion (Dhanikula et al., 1999), PTX:βCD complexes would be internalized thanks to a cholesterol extraction-mediated endocytosis (Rodal et al., 1999; Rosenbaum et al.; 2010). By contrast, conjugated APPZ would be internalized by a receptor-mediated (HER2) endocytosis and this fact possibly would slow down PTX release and would favour its tumor accumulation (Ma et al., 2013; Bhattacharyya et al., 2015; Xu et al., 2013).

4) Co-localisation/uptake studies and mechanisms by CLSM or FACS are also overlooked. Please consider these issues.

As explain at the points 2.7 and 3.3 of the main text, an uptake study of the conjugated APPZ (APPZ-PTX:βCD-T) in the BT474 cell line was performed by means of CLSM (acronym included in line 150). Such breast carcinoma cell line was chosen because of its HER2-overexpression. BT474 cells were cultured and treated with conjugated APPZ and a secondary antibody was employed to analyze HER2 internalization from the cellular membrane to the cytoplasm 4, 24 and 48 hours after treatment. CLSM images were taken also 30 minutes after treatment as control of the membrane HER2 expression. Moreover, an anti-LAMP1 antibody was employed to study if the internalized conjugated APPZ were in the cellular lysosomes 4 (data not shown), 24 (data not shown) and 48 hours after treatment.

As result (Figure 3C and 3D), it was proven that the membrane HER2 was internalized into cells’ cytoplasm after treatment with the conjugated APPZ and that its fluorescent signal co-localized with the LAMP1 one. Thus, it was demonstrated that conjugated APPZ were internalized in HER2-overexpressing cells by endocytosis, thanks to the conjugated-trastuzumab binding to HER2, and that they ended-up in cellular lysosomes.

5) Literature references could be updated.

References [11] (Chakravarthi and Robinson, 2011), [17] (Lazaridis et al., 2008) and [46] (Eskelinen, 2006) have been updated and changed by Chowdhury et al., 2019; Baselga et al., 2014 and Alessandrine et al., 2017, respectively. The other references which are prior to 2010 could not be modified because they have been important or essential for the described work.

Round  2

Reviewer 1 Report

IN my opinion, the revised version is suitable for publication.